# The Role of Sarcopenic Obesity in Cancer and Cardiovascular Disease: A Synthesis of the Evidence on Pathophysiological Aspects and Clinical Implications

**DOI:** 10.3390/ijms22094339

**Published:** 2021-04-21

**Authors:** Erika Aparecida Silveira, Rômulo Roosevelt da Silva Filho, Maria Claudia Bernardes Spexoto, Fahimeh Haghighatdoost, Nizal Sarrafzadegan, Cesar de Oliveira

**Affiliations:** 1Postgraduate Program in Health Sciences, Faculty of Medicine, Federal University of Goiás, Goiânia 74605-050, Brazil; romulordsf@gmail.com; 2Department of Epidemiology & Public Health, Institute of Epidemiology & Health Care University College London, London WC1E 6BT, UK; c.oliveira@ucl.ac.uk; 3Postgraduate Program in Food, Nutrition and Health, Faculty of Health Sciences, Federal University of Grande Dourados, Dourados 79.804-970, Brazil; mariaspexoto@ufgd.edu.br; 4Hypertension Research Center, Cardiovascular Research Institute, Isfahan University of Medical Science, Isfahan 815838899, Iran; f_haghighatdoost@yahoo.com; 5Isfahan Cardiovascular Research Center, Cardiovascular Research Institute, Isfahan University of Medical Sciences, Isfahan 8158388994, Iran; 6Faculty of Medicine, School of Population and Public Health, University of British Columbia, Vancouver, BC V6T 1Z3, Canada

**Keywords:** sarcopenic obesity, obesity, muscle mass, cancer, cardiovascular disease, pathophysiological aspects, mortality

## Abstract

Obesity is globally a serious public health concern and is associated with a high risk of cardiovascular disease (CVD) and various types of cancers. It is important to evaluate various types of obesity, such as visceral and sarcopenic obesity. The evidence on the associated risk of CVD, cancer and sarcopenic obesity, including pathophysiological aspects, occurrence, clinical implications and survival, needs further investigation. Sarcopenic obesity is a relatively new term. It is a clinical condition that primarily affects older adults. There are several endocrine-hormonal, metabolic and lifestyle aspects involved in the occurrence of sarcopenic obesity that affect pathophysiological aspects that, in turn, contribute to CVD and neoplasms. However, there is no available evidence on the role of sarcopenic obesity in the occurrence of CVD and cancer and its pathophysiological interplay. Therefore, this review aims to describe the pathophysiological aspects and the clinical and epidemiological evidence on the role of sarcopenic obesity related to the occurrence and mortality risk of various types of cancer and cardiovascular disease. This literature review highlights the need for further research on sarcopenic obesity to demonstrate the interrelation of these various associations.

## 1. Introduction

Obesity is globally a serious public health concern associated with a high risk of cardiovascular disease (CVD) and various types of cancer [1,2,3]. A recent study that summarized [4] the epidemiological evidence on visceral obesity and its risk for these outcomes clearly demonstrated that it increases the risk of some types of cancer, such as colorectal, pancreatic and gastroesophageal cancer. However, for many types of cancer, the findings remain controversial or are restricted to specific groups like women after menopause [5,6,7,8,9]. Despite the overwhelming body of evidence on the risk of CVD in obese individuals, this association disappears for visceral obesity when the risk is assessed by gender and in older adults [4]. Therefore, there is a clear need for studies that investigate other types of obesity, such as sarcopenic obesity, which increases the risk of CVD and cancer, which are globally the highest causes of mortality, before the COVID-19 pandemic.

Sarcopenic obesity is a relatively new term with several studies under development. Sarcopenia is a condition that affects mainly older adults but also adults and obese individuals. Sarcopenia reduces functionality and increases the risk of falls, fractures, hospitalization and mortality [10,11,12,13,14]. Sarcopenia has been defined by the second European consensus as the presence of low skeletal muscle strength and reduced muscle mass or quality. Sarcopenic obesity is characterized by a reduction in muscle mass and strength occurring simultaneously with excess body adiposity [10].

There are several endocrine-hormonal, metabolic and lifestyle aspects involved in the occurrence of sarcopenic obesity [15,16] that, consequently, affect pathophysiological aspects that can contribute to developing cardiovascular diseases and neoplasms [5,17]. Inflammatory aspects related to both obesity and sarcopenia, such as the unbalanced production of inflammatory cytokines and reactive oxygen species, are also present in CVD and cancer. Therefore, it is important to know the role of sarcopenic obesity in the pathophysiology of these adverse health outcomes.

However, there is no evidence currently showing the role of sarcopenic obesity in the occurrence of CVD and cancer and its potential pathophysiological mechanisms involved. CVD and cancer share many pathophysiological aspects, risk factors, prevention and treatment [18]. The present study is relevant considering the global context of an ever-increasing older adult population, the epidemic and associated burden of CVD and cancer and the increased rates of obesity and sarcopenia, especially among older adults. Therefore, the purpose of this research is to describe the pathophysiological aspects and the clinical and epidemiological evidence on the role of sarcopenic obesity on the occurrence and mortality associated with various types of cancer and cardiovascular disease.

## 2. Definitions and Diagnostic Criteria for Sarcopenic Obesity

Sarcopenic obesity relies on the diagnosis of sarcopenia and obesity. There are different definitions and diagnostic criteria for both conditions, mainly considering adults or older adults and the place where the diagnosis was made, i.e., hospitalized patients or in epidemiological population studies. Thus, the components evaluated in the diagnosis of sarcopenic obesity can vary greatly similar to the combinations of the parameters for sarcopenia [11,19], such as strength, muscle mass and performance and obesity (BMI, waist circumference, body fat percentage, etc.) (Table 1). In addition, each parameter has different methods of measurement with a diversity of cutoff points, such as BMI and muscle mass [20]. Muscle mass, for example, can be estimated by nuclear magnetic resonance (NMR), computed tomography (CT), dual emission X-ray absorptiometry (DXA), electrical bioimpedance (BIA), ultrasonography and even very indirect measurements, such as anthropometric measurements [13,19,21,22]. The most widely used anthropometric measurement to evaluate muscle mass is calf circumference [13,21,23]. Consequently, these differences considerably reduce the possibility of comparisons between studies investigating sarcopenic obesity and its consequences.

At present, there is no consensus for the diagnosis of sarcopenic obesity. Even sarcopenia was formally recognized as a clinically distinct disease only in 2016 [34]. The first consensus on sarcopenia emerged in 2010 [12]. Although some of these consensuses have already been revised, the diagnostic definition of sarcopenic obesity remains open [13,35]. The identification of this health condition, however, is important and can support the implementation of interventions, which can minimize the chances of several negative outcomes on healthy adults and older adults [10], such as functional losses [14], metabolic syndrome [36], respiratory diseases [37], and ultimately, mortality [11]. Several of these outcomes are exacerbated when sarcopenia and obesity occur simultaneously rather than independently [10,19]. Additionally, sarcopenic obesity is associated with a distinct profile of biomarkers, which reflect the reciprocal influence of adipose tissue on skeletal muscle and vice versa [13,19]. It is also important to highlight that obesity can make it difficult to identify low muscle mass in sarcopenic obese individuals. Sarcopenic obesity, therefore, should be addressed as a condition distinct from obesity or sarcopenia.

Due to inconsistencies in the criteria for assessing sarcopenic obesity, its prevalence in population studies can vary from 0 to 100% [19,38]. In the United States of America, the prevalence in individuals aged 60 and older ranged from 3.6% to 94.0%, depending on the diagnostic criteria applied and sex [39]. In Brazil, the multicenter FIBRA study estimated a 4.44% prevalence of sarcopenic obesity in older adults over 65 years, based on BMI and handgrip strength [40]. Thus, the lack of diagnosis consensus of sarcopenic obesity can lead to difficulties in estimating the actual prevalence of this condition. The variable most frequently associated with sarcopenic obesity is age over 80 years [19].

## 3. Pathophysiology of the Association of Sarcopenic Obesity, Cardiovascular Disease and Cancer

Multiple pathophysiological changes occur due to sarcopenic obesity, like an excess of proinflammatory cytokines from exacerbated adipose tissue, altered expression of adipokines by adipose cells and a high flow of lipids to muscle fibers [17,19]. The atrophy of skeletal muscle cells decreases the expression of GLUT4 in muscle fibers and reduces the demand for insulin-dependent glucose uptake [41]. The proinflammatory state and lipid accumulation in muscle fibers, on the other hand, induce the activity of intracellular kinases that phosphorylate and inactivate the insulin receptor and its substrates [17]. Insulin resistance is a central mechanism for the association of sarcopenic obesity with both cardiovascular disease and cancer.

### 3.1. Cardiovascular Disease

Sarcopenic obesity can reduce insulin sensitivity leading to hyperglycemia, which induces increased insulin secretion by the pancreas and hyperinsulinemia [42]. High levels of insulin, in turn, sensitize smooth vascular cells to proliferate via the MAPK pathway [43] and interfere with the control of blood pressure exercised by the renin–angiotensin–aldosterone system [44]. Smooth muscle cells thus proliferate and increase the resistance of the vascular musculature. Angiotensin II and aldosterone, in turn, act on their endothelial receptors and also on the heart muscle, reducing the production of nitric oxide and activating myocardial tissue remodeling pathways [17]. Such processes likely lead to high blood pressure, myocardial fibrosis and cardiac stiffness [17].

Hyperglycemia due to sarcopenic obesity causes the glycosylation of several circulating macromolecules, such as albumin, lipoproteins, insulin and hemoglobin [45]. The so-called advanced glycation end products (AGEs) interact with cell surface receptors for advanced glycation end products (RAGEs), and such phenomenon is associated with the progression of several chronic diseases [46]. The interaction of these proteins with cellular receptors in the endothelium leads to the activation of the transcription factor NFkB and to the production of reactive oxygen species (ROS) in cells. The damage caused by this stress can include endothelial dysfunction, arterial stiffness and microvascular damage [17]. AGEs are also involved in atherosclerosis and coronary diseases once they initiate oxidation of low-density lipoproteins (LDL) via endothelial oxidative stress [46,47]. Another harmful effect of AGEs is the induction of cardiomyocyte apoptosis after infarction by activation of ERK1/2 and p53 pathways [48].

Sarcopenia and obesity are also involved in impairing the autonomic nervous control of cardiac function. The exacerbation of the renin–angiotensin–aldosterone system acts on the hypothalamic paraventricular nucleus, causing excitation of the sympathetic tone [49]. On the other hand, the reduction in muscle mass also reduces the expression of myokines, including irisin, involved in the maintenance of vagal tone and in the parasympathetic modulation of cardiac function [50,51].

In the muscles, insulin resistance, accumulation of adipose tissue and inflammation exacerbate the production of reactive oxygen species (ROS) by mitochondria, causing mitochondrial damage, activating proteolytic intracellular pathways and inducing apoptosis [17]. Insulin resistance reduces the influx of glucose into the muscle fiber, increasing using fatty acids as an energy substrate. The high yield of fatty acids in ATP production, however, increases the ATP/ADP ratio in the mitochondria, reducing the activity of the electron transport chain, which results in greater production of ROS s [52]. Cytokines and peri muscular lipid accumulation induce the activity of transcription factors associated with proteolysis and apoptosis, such as FoxO and NFkB, exacerbating the damage to muscle fiber [53,54]. This scenario, defined as lipotoxicity, affects muscle contractility, which can lead to the death of cardiomyocytes and heart failure [17]. Hyperlipidemia, inflammation and ROS are also directly involved in the genesis of atherosclerosis when oxidized LDL molecules enter the vascular intima and are phagocytized by macrophages. These macrophages accumulate cholesterol in their membranes and deposit themselves under the endothelium, undergoing apoptosis and causing the accumulation of collagen fibers in fibroatheroma plaques [17,55].

Sarcopenic obesity and cardiovascular disease, therefore, share common etiological mechanisms (Figure 1). Body composition, inflammation, insulin resistance and oxidative stress produce systemic repercussions that greatly impact the heart and blood vessels. Among the main consequences of this imbalance are hypertension, dyslipidemia, metabolic syndrome, type 2 diabetes mellitus, non-alcoholic liver steatosis, heart failure and major cardiovascular events [17,56].

### 3.2. Neoplasms

Neoplasms also share etiological mechanisms with sarcopenic obesity. As described above, loss of muscle mass and obesity induce insulin resistance, hyperinsulinemia and hyperglycemia. Oxidative stress and lipotoxicity generated by insulin resistance and inflammation produce potentially mutagenic ROS in protooncogenes and tumor suppressor genes [57]. Another consequence of insulin resistance is the reduction in the uptake of amino acids by muscle cells, altering the balance of protein synthesis/degradation in favor of proteolysis. It is proposed that the amino acids released by muscle proteolysis in sarcopenia contribute to the supply of tumor growth [58].

The excess of circulating insulin, in sarcopenic obesity, induces growth and proliferation of some cell lines, favoring the appearance of neoplasms [57]. Tumor cells have a high expression of the insulin receptor, necessary for the uptake of glucose used in the generation of energy and cell division. Hyperinsulinemia also favors the cross-interaction of insulin with the IGF-1 receptor and vice versa, another route by which tumor growth can occur. In addition, sarcopenia, sarcopenic obesity and diabetes mellitus are associated with gastric carcinogenesis and the appearance of pre-cancerous gastric lesions [57,59].

Hyperglycemia, caused by insulin resistance in sarcopenic obesity, favors the energy metabolism of neoplastic cells by the Warburg effect [42]. High glycemia also acts on oncogenesis through the interaction of glycated proteins with tissues. AGE-RAGE interactions promote tumor initiation (via enhanced early neoplastic cell survival) [60]; progression (via malignant cell line proliferation) [61]; invasion (via enhanced activity of matrix metalloproteinases) [62]; and metastasis (via angiogenesis) [46,63,64].

The systemic inflammatory condition in sarcopenic obesity is also involved in oncogenesis. Proinflammatory cytokines stimulate insulin resistance, proteolysis and the production of ROS, the effects of which were mentioned above. Hypertrophy and hyperplasia of adipocytes caused by obesity lead to disordered growth of white adipose tissue, being accompanied by hypoxia and cell death in less vascularized parts of this tissue [65]. This tissue damage is succeeded by releasing molecular patterns derived from cellular damage (DAMPs), which activate several cells of innate immunity [66]. Some macrophages organize themselves into crown-like structures (CLS) around the adipocytes and express an inflammatory pattern intermediate to the classic patterns M1 and M2 [65,67]. Macrophages in CLS recruit neutrophils and mast cells to adipose tissue and also stimulate stromal cell differentiation into myofibroblasts. Elastases produced by neutrophils and cathepsin S produced in mast cells are involved in promoting the progression of tumor cells. Myofibroblasts and hypertrophied adipocytes produce type VI collagen, altering tissue mechano-homeostasis and also releasing VEGF, inducing neoangiogenesis. Macrophages also stimulate the expression of apoptosis-inducing membrane receptors (PD-1) in CD8 + T cells, reducing the cytotoxic response against tumor cells. The presence of CLS in the adipose tissue of several organs is a biomarker indicative of a worse prognosis, as in the case of prostate and breast neoplasms [65]. Thus, the inflammation of sarcopenic obesity is associated with the occurrence of neoplasms by three main mechanisms: sustained stimulation of cell proliferation, neoangiogenesis and reduced anti-neoplastic immune response.

High adiposity, characteristic of obesity, also participates in oncogenesis. As discussed, the growth of visceral and ectopic adipose tissue participates in inflammation and insulin resistance. However, fat cells and lipids still have some more ways to contribute to cancer. Fatty acids are converted to acetyl coenzyme A by the endothelium of the lymphatic vessels, which causes epigenetic changes critical for lymphangiogenesis [68], an important metastasis pathway. Cholesterol metabolites interact with estrogen receptors or glucocorticoid receptors in different types of breast cancers, promoting tumor progression [69,70]. Hypertrophied adipocytes release chemokines that attract neoplastic cells in periprostatic adipose tissue and also send lipids to modulate the metabolic pattern of melanoma cells, inducing their growth and invasion [71].

The increase in BMI, present in sarcopenic obesity, is correlated with higher levels of estrone, estradiol and testosterone [57]. These hormones are stimulators of some tumor cell lines. Sex hormone-responsive neoplasms are strongly associated with obesity and sex hormone levels. As previously discussed, amino acids derived from proteolysis and sarcopenic condition may serve as an additional supply to tumor nutrition. Thus, sarcopenic obesity is associated with endometrial, breast, uterine, ovarian and prostate cancers [18]. The mechanism of action of sex hormones in these diseases involves the participation of the aromatase enzyme, whose expression is increased with the increase of adipose tissue. Thus, the circulation of estradiol is increased, which can act as a tumor inducer. Estradiol promotes cell survival and proliferation through the transcription of oncogenes [72]. This hormone also interacts with pathways of cell growth factors, such as EGF, IGF and FGF [73].

The altered expression of adipokines and myokines can also be associated with oncogenesis. Elevated leptin levels are associated with the emergence of gastric, colorectal, breast, liver, prostate cancers and acute lymphoblastic leukemia [65,74]. This adipokine promotes proliferation in neoplastic cells by activating the Akt/mTOR and JAK/STAT intracellular pathways. The chronic elevation of its levels causes resistance to leptin, which reduces lipolysis and favors even more adipose accumulation, inflammation, oxidative stress and insulin resistance [74]. Adiponectin, at reduced levels in obesity, consequently has less control over cell proliferation. Low adiponectin levels also allow greater infiltration of inflammatory cells in adipose tissue, facilitating the formation of CLS, inflammation and insulin resistance [65,74]. IL-15 and irisin, two myokines stimulated by physical exercise, have their levels altered in obesity and sarcopenia [74]. Deficiency in these myokines impairs the stimulation of lipolysis, muscle hypertrophy and the maintenance of homeostatic levels of leptin and adiponectin [74]. Irisin still controls the proliferation, migration and viability of tumor cells in different tissues and by different intracellular pathways [75].

Sarcopenic obesity and oncogenesis, therefore, are mediated by mechanisms, such as insulin resistance, type 2 diabetes, inflammation, immune damage, imbalance of adipokines and myokines, and remodeling of the extracellular matrix and the vasculature. Together, these phenomena correlate with the incidence of several types of cancer, including liver, pancreas, stomach, colorectal, kidney, bladder, endometrium, prostate, breast, non-Hodgkin’s lymphomas and leukemia [57,59,65,76].

## 4. Epidemiological and Clinical Evidence of the Association between Sarcopenic Obesity and Cardiovascular Disease

Excess fat mass, especially visceral fat mass, is a well-known risk factor for cardiovascular disease (CVD). In contrast, greater muscle mass is considered to be cardioprotective through its myokines [77]. Therefore, higher CVD risk in sarcopenic obese individuals can be expected [78]. However, it has been shown that older adults with greater body mass index (BMI) are less likely to die from CVD, which is called the “obesity paradox” [79,80,81]. It seems that the favorable effects of obesity on CVD prognosis are meditated through greater lean mass (LM). It is also interesting to note that high-fat mass besides increased LM may play a protective role against CVD, especially in those with low-grade systematic inflammation. In other words, high-fat mass may not be deleterious in the absence of systemic inflammation (hsCRP > 3 mg/dL) [82]. However, it seems that this paradox is more evident in chronic heart failure and coronary heart disease patients [81]. In addition, since a decline in skeletal muscle mass and function adversely affect health-related quality of life in older adults, obesity cannot effectively reduce the risk of chronic diseases in sarcopenic elderly individuals [81]. Therefore, this paradox highlights the relevance of exploring the possible association between sarcopenic obesity and CVD.

Sarcopenic obesity may adversely affect metabolic profile [78,83]. In the Korean National Health Examination and Nutrition Survey (KNHANES), various metabolic abnormalities that are associated with CVD, such as vitamin D deficiency, elevated serum insulin, triglyceride and ferritin levels, low serum HDL-C, insulin resistance and metabolic syndrome, were more frequent in sarcopenic obese subjects [84]. These risk factors were similarly prevalent in male and female sarcopenic obese individuals. However, systolic blood pressure was higher only among sarcopenic obese men, not women, than non-sarcopenic non-obese participants [84]. Consistently, in sarcopenic obese patients, who undergo peritoneal dialysis, serum levels of triglyceride and hs-CRP were higher than other counterpart groups, while no difference was found for other cardiometabolic risk factors like serum lipids and inflammatory markers [85].

To date, several observational studies have been conducted to explore the association between sarcopenic obesity and CVD. However, the existing evidence remains inconclusive (Table 2). The discrepancies between studies could be attributable to the variations between study design and adjusted variables (i.e., weight vs. height^2^), the methods used to measure muscle mass, body composition, and fat mass, the methods and cutoff points used to define sarcopenia (i.e., muscle mass, muscle strength, or both) and obesity (visceral vs. general adiposity), as well as the statistical methods used to assess the risk. These factors, therefore, lead to a non-standardized definition for sarcopenic obesity and, consequently, provide a great heterogeneity between studies [78,86]. Many studies have shown that visceral adiposity is a better predictor of CVD than general obesity [87,88]. The study conducted by Fukuda et al. is a good example to highlight the role of sarcopenic obesity definition in their observed associations [88]. In this retrospective study that was conducted on 716 diabetic patients with a mean age of 65 years, the relationship of sarcopenia in combination with four different patterns of obesity with the risk of CVD occurrence was assessed [88]. They measured body composition using whole-body dual-energy X-ray absorptiometry (DXA) [88]. After a follow-up period of 2.6 years, it was found that only sarcopenic individuals with either high android to gynoid fat ratio or with high android fat mass were at greater risk of CVD, whereas a high percentage of body fat and BMI ≥25 kg/m^2^ were not related to the risk of CVD occurrence [88]. Android fat (or central fat) and gynoid fat (or peripheral fat) form truncal fat, which is responsible for fat depositions in the upper and lower areas of the body [89]. In support of this finding, a large population-based prospective cohort study, the Elderly Nutrition and Health Survey in Taiwan [87], demonstrated that the coexistence of sarcopenia and elevated triglyceridein abdominal obese people was associated with a higher risk of CVD death, independent of their BMI level [87]. Similarly, in the Korea National Health and Nutrition Examination Survey, cancer-survivors and noncancer participants with sarcopenic abdominal obesity were at three- and four-times greater risk for having ≥10% CVD risk score compared with non-sarcopenic, non-obese counterparts [90]. Examining the association between first CVD events and CVD mortality in relation to sarcopenic obesity (defined as low handgrip strength and BMI ≥ 30 kg/m^2^) using data from the UK biobank illustrated that sarcopenic obese, irrespective of their CVD history, had the greatest odds for coronary heart disease occurrence or death [91], while for CVD events and mortality, sarcopenic obesity only increased the risk in those with no history of CVD. However, when the investigators used waist to hip ratio (WHR) to measure central adiposity, the associations became significant even in participants with CVD history [91]. The results from the US National Health and Nutrition Examination Survey (NHANES), including 11,317 adults, also demonstrated that sarcopenic obesity increased the risk of CVD by over eight times in both metabolically healthy and unhealthy subjects [92].

In contrast to the findings from the above-mentioned studies, some studies failed to find any association between sarcopenic obesity and CVD. A cross-sectional study that was conducted with 99 acute myocardial infarction patients with a mean age of 71.6 years illustrated that sarcopenic obesity (defined based on muscle strength, physical performance and muscle mass measured by bioelectrical impedance analysis and WC) affected approximately one-third of patients, but was not correlated with thrombolysis [93]. After an 11 year follow-up period using data from the British Regional Heart Study, conducted in old men aged 60–79 years, no significant association was found between sarcopenic-abdominal obesity and coronary heart disease events. Nevertheless, in terms of CVD mortality, their results suggested a significant, independent direct association in individuals with sarcopenic abdominal obesity compared to non-sarcopenic and non- abdominally obese subjects (HR = 1.37, 95% CI: 1.02, 1.82) [30]. Their results also suggested no significant interaction between obesity and sarcopenia in relation to any of the outcomes [30].

Sarcopenic obesity may also adversely affect clinical prognosis in patients with CVD. There is some evidence suggesting that in patients, who underwent cardiovascular surgery, after adjustment for potential confounders, sarcopenic obesity (defined as low muscle attenuation and abdominal adiposity) was associated with poorer muscle function and three times higher risk of mortality after cardiovascular surgery [94].

## 5. Epidemiological and Clinical Evidence of the Association between Sarcopenic Obesity and Cancer

The impact of sarcopenic obesity (SO) on cancer patients is still unclear in the literature, with a limited number of studies that have investigated the implications of different types of cancer [95,96,97,98]. Most research has focused on the risk of occurrence, impacts on treatment and mortality. With regard to the different types of cancer, there are studies looking into obesity, abdominal obesity [99,100,101,102] or sarcopenia separately [103,104], but not sarcopenic obesity, which is the focus here.

It is worth mentioning that there are two integrative reviews on the impacts and clinical implications of sarcopenic obesity in cancer [6,7] published in 2016 and 2018. These reviews, however, included only the evidence available until 2017. Our review takes a broader approach, including evidence published until March 2021. We highlight the available evidence on the prevalence of sarcopenic obesity and its impact on cancer incidence, possible treatment complications and patient survival.

### 5.1. Occurrence and Association of Sarcopenic Obesity in the Occurrence of Cancer

The sarcopenic obese phenotype has been increasingly identified among cancer patients [105,106,107], which is probably due to the increasing prevalence of obesity all over the world combined with intense muscular catabolism caused by the treatment and/or by the staging of the disease itself [108]. It is worth mentioning that obesity has reached epidemic proportions worldwide and that this, in turn, increases the risk of several types of cancer [1,18].

Sarcopenic obesity is more common in older adults (≥50 years old) [109], and the prevalence in individuals with cancer can vary substantially between 1% and 41.7%, as shown in Table 3. This range of prevalence in several studies can be related to the different criteria used for the diagnosis of sarcopenic obesity, to the various types of cancer and their staging, aspects that limit understanding the panorama of occurrence and reveals the need for more research with other types of cancer not yet investigated. Further details on the studies that investigate the prevalence of sarcopenic obesity in different types of cancer according to staging, age and criteria used to define sarcopenic obesity are presented in Table 3. For the synthesis of this evidence, we opted for studies with more than 60 individuals.

Regarding tumors of the digestive system, the occurrence of sarcopenic obesity ranged from 2% (colorectal cancer, *n* = 259) [110] to 41.7% (gastric cancer and *n* = 8356) [59]. For tumors in the endocrine system, only five studies were found in which the prevalence ranged from 7% to 25% in a survey of 228 men with pancreas cancer. In the genital and urinary system, we observed two studies that were performed on older adult men, one on bladder cancer and a prevalence of sarcopenic obesity of 4.5% and another on prostate cancer in which sarcopenic obesity prevalence was 12.6% [119]. Three studies [8,9,95] in women with breast cancer and the prevalence of sarcopenic obesity ranged from 1 to 7.2. The higher prevalence of OS seems to be more related to gastric and pancreas cancers.

A cohort study of 8356 Korean adults who underwent gastroduodenoscopy at a screening center concluded that sarcopenic obesity might be a risk factor for gastric carcinogenesis. Therefore, sarcopenic obesity is significantly associated with the diagnosis of gastric cancer [59]. Sarcopenic obesity was significantly associated with gastric cancer.

### 5.2. Sarcopenic Obesity in the Complications of the Most Common Anticancer Treatments and Other Clinical Implications

We tried to synthesize the available evidence investigating the clinical implications and cancer survival that is probably affected by sarcopenic obesity. Only one study reported a reduction in functionality and another reduction of handgrip strength associated with sarcopenic obesity, both in male patients [105,106]. Further studies are needed to assess the impact of sarcopenic obesity in cancer patients by identifying whether the impact is the same as in sarcopenic individuals or whether the synergy of occurrence of sarcopenic obesity aggravates variables of functionality and even frailty in older adults.

Regarding the clinical and surgical complications of cancer patients and their possible association and worsening in patients with sarcopenic obesity, we found 12 studies that reported an association of sarcopenic obesity with postoperative complications [32,110,113], toxicity to chemotherapy [96,112], longer hospital stays or readmission [24,110], and higher complication rate [97]. However, two studies differed from the previous studies. One found that sarcopenic obesity did not impact on worst post-surgical results in patients with gastric cancer [98], and another reported that sarcopenic obesity was not a predictor of toxicity to chemotherapy in prostate cancer [119].

It has been stated that sarcopenic obesity adversely impacts treatment tolerance, long-term expectations and is an important indicator of adverse outcomes and a prognostic factor for complications in cancer patients [109]. In the present study, we concluded that there is still a very limited number of studies evaluating the impacts of sarcopenic obesity on several clinical and treatment variables of different types of cancer. These few studies are described below.

A retrospective study with 198 gastric cancer patients, median age 73.5 years, who underwent gastrectomy, concluded that sarcopenic obesity was not associated with worse outcomes after gastric cancer surgery [98]. Of the total of 206 patients with gastric cancer, who were overweight or obese, they identified that sarcopenia had a significant association with a six times greater risk of postoperative complications (OR 6.07; 95% CI = 1.904–13.359; *p* = 0.002), hospital costs more elevated (*p* = 0.003) and higher readmission rate in 30 days after gastrectomy (*p* = 0.035) compared to non-sarcopenic [120]. A cohort of patients diagnosed with advanced gastric cancer receiving chemotherapy, all subjects with sarcopenic obesity (10% of the followed-up population) ended treatment early due to toxicity, that is, unwanted adverse effects resulting from treatment [96]. Sarcopenic obese patients with gastric cancer and undergoing radical gastrectomy, along with age, combined open surgery and resection, have a higher risk of severe postoperative complications than all other patients. The risks identified were obesity and sarcopenic vs. normal, OR = 6.575 *p* = 0.001; sarcopenic obesity vs. obesity, OR = 5.833 *p* = 0.001; sarcopenic obesity vs. sarcopenia, OR = 2.571 *p* = 0.032) [32]. Sarcopenic obesity is a predictor of severe postoperative complications (gastrectomy) in patients with gastric cancer. Postoperative complications promote significantly longer hospitalizations, incur higher hospital costs and increase health costs [32].

Low lean body mass, mostly muscle, can represent a modifiable risk factor in patients with cancer, especially in non-advanced stages, and should be considered [121]. Patients with obesity and low muscle mass had more complications in the same period compared to non-obese patients with low muscle mass (22.0% vs. 13.0%; *p* = 0.019) [113]. Sarcopenic obesity did not worsen disease-free survival rates and complications after partial liver resection for colorectal liver metastasis [24].

Data from a clinical trial showed that patients with resectable esophageal cancer and sarcopenic obesity had a significantly higher risk of toxicity (OR 5.54; 95% CI = 1.12–27.44) compared to non-sarcopenic obese patients [112]. In contrast, in metastatic prostate cancer resistant to castration, sarcopenic obesity was not predictive of toxicity associated with chemotherapy (*p* = 0.511), but the authors clarify that the prognostic value of sarcopenic obesity in the study is unclear and justify the small number of patients [119].

Sarcopenic obesity was considered an adverse factor for women with surgical breast cancer. Sarcopenic patients with high body mass index (BMI) had a worse prognosis than those with normal BMI [95]. Older age, diabetes, stroke, heart failure, cancer and kidney disease predict a worse prognosis in both individuals with “low lean mass” and those with “low lean mass and obesity” [122].

Evidence displayed in Table 4 mostly suggests that sarcopenic obesity is associated with worse clinical and post-surgical outcomes. However, when one takes into account the severity and the number of cancer cases globally and their associated mortality risk, it is worrying that there are only a few studies that investigated how sarcopenic obesity is involved in these different outcomes. Therefore, more research is needed on different types of cancer evaluating the impact of sarcopenic obesity since they may potentially elucidate several issues not yet clarified and point to cancer treatments that prevent or reduce the impact of sarcopenic obesity.

### 5.3. Impact of Sarcopenic Obesity on Cancer Mortality

Sarcopenic obesity is an important prognostic factor for survival in cancer patients [109]. We found 12 studies that assessed the potential impact of sarcopenic obesity on cancer patient’s survival. Ten studies observed reduced survival [8,96,97,105,111,114,115,116,118,123], and three studies did not find a significant association [9,24,124].

Out of the nine studies that evaluated the impact of sarcopenic obesity on digestive system cancer mortality, four were carried out with patients with pancreatic cancer. A study of 228 patients at the University of Nottingham hospitals diagnosed with unresectable pancreatic cancer found that those with sarcopenic obesity had a significant reduction in survival (*p* = 0.013). The authors attributed this association to the presence of lipid infiltration in the inter-myocellular spaces of sarcopenic obese individuals [116]. The combination of visceral obesity and sarcopenia was the greatest predictor of postoperative death in patients undergoing pancreaticoduodenectomy for resectable pancreatic cancer and found a high risk of mortality 60 days after the surgical procedure (odds ratio (OR) 6.76 95% CI, 2.41 to 18.99; *p* < 0.001) [125]. Sarcopenic obese patients undergoing resection for resectable ductal pancreatic adenocarcinoma have significantly shorter overall survival and higher complication rates [97]. After resection of pancreatic cancer of 301 patients in retrospective observation, it was identified that sarcopenic obesity was associated with mortality and still with a very poor prognosis among OS patients [118].

A study with 250 patients, mean age 63.9 ± 10.4 years, with tumors in the gastrointestinal (colorectal) or respiratory tract and obese (34.3 ± 4.4 kg/m^2^) identified an association between sarcopenic obesity and risk of mortality for cancer. Sarcopenic obesity was an independent predictor of survival (risk ratio (RR) 4.2, 95% CI 2.4–7.2; *p* < 0.0001) [105]. In this same study, sarcopenic obesity was more prevalent in males (*p* = 0.013), in those who had colorectal (gastrointestinal) cancer compared to other cancer sites (*p* = 0.019), and in patients aged 65 years or over compared to younger patients (*p* = 0.08) [105]. A study with 1384 patients with nonmetastatic colorectal cancer who received surgical treatment identified a negative association between sarcopenic obesity with overall survival (hazard ratio (HR) 1.395, 95% CI 1.067–1.822; *p* = 0.015). Multivariate analysis found that being a woman was a protective factor (hazard ratio (HR) 0.787, 95% CI 0.624–0.992; *p* = 0.043) [114]. Another study involving 805 patients, between 61 and 77 years of age, with colon cancer undergoing elective surgery for resection, showed higher mortality within 30 days (*p* < 0.001) of the surgery among individuals with obesity and low muscle mass [113]. Another study found divergent results in which sarcopenic obesity did not worsen overall survival rates (HR: 0.663, 95% CI: 0.386–1.141; *p* = 0.138)) after partial liver resection for colorectal liver metastasis in 171 adult-elderly subjects, median 64 years (24–86) [24]. Regarding advanced gastric cancer, sarcopenic obesity was associated with shorter survival in 48 patients in the retrospective analysis [96], but because it is a small sample, the statistical inference may be compromised.

Preoperative sarcopenic obesity was an independent risk factor for death and recurrence of hepatocellular carcinoma in individuals undergoing primary hepatectomy [117]. The authors retrospectively analyzed data from 465 patients undergoing hepatectomy and found that when comparing patients with non-sarcopenic obesity, patients with sarcopenic obesity had worse survival and worse recurrence-free survival. Multivariate analysis identified sarcopenic obesity as a significant risk factor for death (RR = 2.50, 95% CI = 1.336–4.499; *p* = 0.005) and hepatocellular carcinoma recurrence (RR = 2.031, 95% CI = 1.233–3.222; *p* = 0.006) after hepatectomy [117].

An integrative review that summarized the mechanisms of the negative association of changes in body composition with the risk and treatment of breast cancer concluded that insulin resistance, chronic inflammation and other metabolic factors are conditions observed in patients with breast cancer, who have sarcopenic obesity, and this increases the risk of death [5]. A study of 3241 patients with nonmetastatic breast cancer showed that sarcopenia is poorly recognized and is associated with a significant increase in the risk of death. Muscle mass and adipose tissue are more strongly associated with survival than BMI, suggesting that the assessment of these tissues would be more useful in identifying women at risk of low survival. Sarcopenia and adiposity are important risk factors and should be considered together when assessing the risk and prognosis of this population [126]. A study that evaluated the association between skeletal muscle and overall survival of 166 patients with metastatic breast cancer undergoing palliative chemotherapy found that there was no significant association in women with sarcopenic obesity [9].

Although obesity is considered an important risk factor for developing various types of cancer, the presence of obesity seems to be a paradoxical protective factor that can improve response to treatment and, consequently, survival in patients with several chronic diseases, but there are conflicting and controversial results [127]. A possible explanation for this paradox could be the diagnosis of obesity based on the determination of BMI (in kg/m^2^), as this indicator does not accurately differentiate each tissue individually, such as lean mass and fat mass [128,129]. Obese individuals, with or without cancer, who exercise more and are not insulin resistant or hypertensive, could then be at lower risk for mortality.

Most studies suggest that sarcopenic obesity is associated with lower survival rates in cancer patients. However, around 56% of these studies were conducted on pancreatic cancer, followed by esophageal and breast cancer. It is likely that more studies had investigated pancreatic cancer due to its higher mortality rate, especially in late diagnosis [130,131]. Additional research is needed to assess the impact of sarcopenic obesity on the survival rate from several types of cancer. This will, in turn, help the development and implementation of more accurate and appropriate strategies in clinical practice to promote more assertive cancer treatments.

## 6. Conclusions

Sarcopenic obesity has become more important in recent years. One of the main reasons for this growing research interest is the ever-increasing global population aging associated with chronic conditions, such as CVD and cancer, which share common aspects. Sarcopenic obesity not only increases the occurrence but also morbidity and death rates of CVD and cancer, which may be caused by similar pathophysiological pathways. Its prevalence may be underestimated due to the lack of standard methods and definitions used in previous studies. Specific preventive strategies, such as caloric restriction associated with protein supplementation and resistance exercise, are necessary to target sarcopenic obesity in older adults to prevent CVD, cancers and other adverse health conditions.

## Figures and Tables

**Figure 1 ijms-22-04339-f001:**
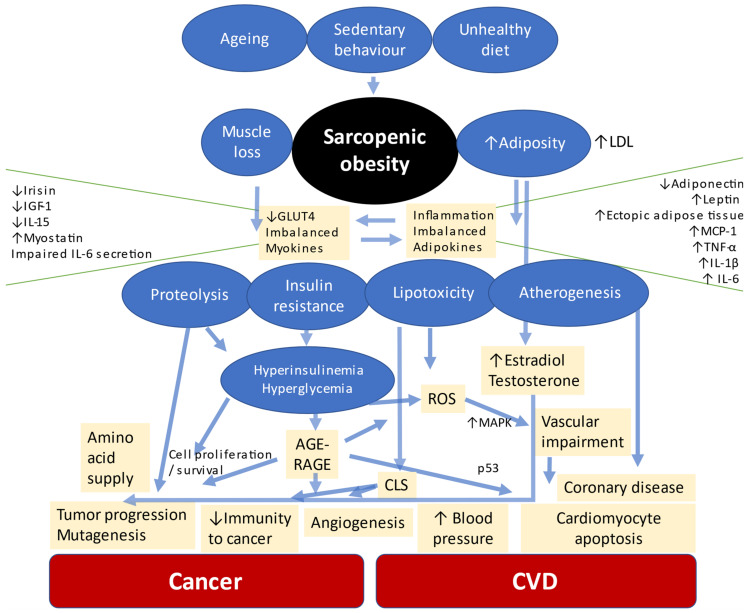
Main pathophysiological mechanisms shared in developing CVD and cancer in individuals with sarcopenic obesity. AGEs = advanced glycation end products; CLS = crown-like structures; CVD = cardio-vascular disease; GLUT4 = glucose transporter type 4; IGF-1 = insulin-like growth factor 1; IL = interleukin; MAPK = mitogen-activated protein kinase; MCP-1 = monocyte chemoattractant protein 1; p53 = tumor protein P53; RAGE = receptor of advanced glycation end products; ROS = reactive oxygen species; TNF-α = tumor necrosis factor α.

**Table 1 ijms-22-04339-t001:** Different components and methods are used for the diagnosis of sarcopenic obesity (muscle mass, muscle strength, physical performance and several types of obesity).

Muscle or Lean Mass	Muscle Strength	Physical Performance	Obesity/Adiposity
CT [24,25]	Handgrip strength [26,27]	Gait speed [13,28]	BMI [25,26]
DXA [14,29]	5 times stand test [13]	Timed up-and-go test [13]	% Body fat [14,29]
BIA [29,30]		Short physical performance battery [13]	Waist circumference [30,31]
Ultrasonography [22]		400-m walk test [13]	Visceral fat area [32,33]
Anthropometry [21,30]			Fat mass index [8,30]

BIA = bioelectrical impedance analysis; BMI = body mass index; CT = computerized tomography; DXA = dual-energy X-ray absorptiometry.

**Table 2 ijms-22-04339-t002:** Association of sarcopenic obesity with cardiovascular diseases and mortality.

Author/Year	Setting/Population	Outcome(s)	Sarcopenic Obesity	Adjustment for	Association Sarcopenic Obesity and CVD	Summary
		Cross-Sectional Studies				
[84]	Korean National Health Examination and Nutrition Survey (KNHANES)/2943 subjects (1250 men and 1693 women) 60 years + older	Cardiometabolic risk factors, metabolic syndrome	Appendicular skeletal muscle mass (ASM) divided by weight BMI ≥ 25 kg/m^2^	-	MetS prevalence Sarcopenic obese Men: 60.9% Women: 71.1% Non-sarcopenic, non-obese Men: 11.6 Women: 32.4	Sarcopenic obesity was more closely associated with insulin resistance, metabolic syndrome, and cardiovascular disease (CVD) risk factors than any other group
[85]	79 peritoneal dialysis patients	CVD risk factors	Low SMM plus low muscle function Percentage of total body fat	-	mean ± SD Hs-CRP Sarcopenic obese: 8.0 ± 0.3 non-sarcopenic obese: 6.8 ± 0.7 Triglyceride Sarcopenic obese: 234.0 ± 61.0 non-obese sarcopenic: 186.0 ± 28.0 non-sarcopenic obese: 185.0 ± 30.0	Sarcopenic obese subjects had higher levels of hs-CRP and triglyceride compared with non-sarcopenic obese subjects No significant difference was found for total cholesterol, LDL, HDL, LPa, MDA, and sICAMI
[93]	99 older adults patients with acute myocardial infarction	Cardiovascular risk and prognostic markers	Loss of muscle mass, loss of muscle strength and poor physical performance Abdominal obesity using sex-specific WC cutoff points	-	Median (IQR) Thrombolysis in myocardial infarction Sarcopenic obese: 4.0 (2.0–6.0) Creatine kinase MB isoenzyme Sarcopenic obese: 4.2 (1.4–17.4) Troponin Sarcopenic obese: 0.06 (0.02–0.86)	Sarcopenic obesity affected approximately one-third of patients and was not associated with any of the prognostic predictors
[92]	National Health and Nutrition Examination Survey 25,270 adults	CVD	Sarcopenia sex-specific ASMI cutoff points based on the revised European consensus BMI ≥ 25 kg/m^2^	Age, sex, ethnicity, and smoking status	OR (95% CI): Sarcopenic obese, metabolically healthy: 8.59 (2.63, 28.04) Sarcopenic obese, metabolically unhealthy: 8.12 (4.04, 16.32)	Sarcopenic obesity was associated with increased risk of CVD
			**Cohort Studies**			
[30]	British Regional Heart Study/4252 men aged 60–79 years 11.3 years follow-up	CHD events CVD events	Low-fat free mass index (FFMI) High-percentile points of fat mass index (FMI) corresponding to the WC obesity	Age, smoking, alcohol, occupational social class, physical activity	HR (95% CI) CHD events 1.13 (0.82–1.56) CVD events 1.01 (0.79–1.29)	No significant association for any outcome
[88]	716 Japanese diabetic patients (mean age 65 ± 13 years; 47.0% female) 2.6 years follow-up	CVD	The coexistence of low SMI and obesity determined by android to gynoid ratio	HDL cholesterol, HbA1c, eGFR, use ACEIs or ARBs, use DPP4 inhibitors, history of CVD	HR (95% CI) Sarcopenic obesity: 2.63 (1.10–6.28)	Sarcopenic obesity was a predictor of incident CVD
[91]	UK Biobank/452 931 men and women aged 40 and 69 years 5.1 y follow-up	CHD events and CVD events and	Handgrip strength Obesity = BMI > 0 Obesity = WHR ≥ 0.95 in men and ≥ 0.80 in women	Age, sex, ethnicity, smoking, alcohol consumption, diabetes, physical activity, deprivation	HR (95% CI) Sarcopenic obesity (BMI) CHD events NO history: 1.79 (1.54, 2.08) History: 1.52 (1.37, 1.70) CVD events NO history: 1.24 (1.11, 1.38) History: 1.10 (0.97, 1.24) Sarcopenic obesity (WHR) CVD events NO history: 1.30 (1.22, 1.39) History: 1.24 (1.15, 1.34)	Sarcopenic obese subjects greater risk of CHD events and mortality irrespective of their CVD history Regarding CVD events and mortality, there was a positive association only in those who did not have a history of CVD
[90]	Korea National Health and Nutrition Examination Survey/19,019 individuals 10 y follow-up	10-year CVD risk scores based on Framingham risk model	handgrip strength, BMI	Sociodemographic dietary intake time cancer diagnosis current cancer therapy (in cancer survivors)	OR (95% CI) Cancer survivors 3.61 (1.60, 8.13) Noncancer participants 4.24 (3.44, 5.22)	Cancer survivors and noncancer with sarcopenic obesity had increased 10-year CVD risk scores
		**Mortality–Cohort Studies**				
[30]	British Regional Heart Study/4252 men aged 60–79 years 11.3 years follow-up	CVD mortality	Low-fat free mass index (FFMI) and high percentile point of fat mass index (FMI) corresponding to the WC obesity	Age, smoking, alcohol, occupational social class, physical activity	HR (95% CI) CVD mortality 1.11 (0.81–1.53)	No significant association
[87]	Elderly Nutrition and Health Survey/1485 elderly individuals aged over 65 years 9.2 years follow-up	Cardiovascular mortality	Low skeletal muscle mass index and abdominal obesity plus hypertriglyceridemia (>150 mg/dL)	Age, gender, BMI, high blood pressure, low HDLC, included hs-CRP, eGFR, comorbidities, smoking, drinking, physical activity	11.3% (n = 168) CVD death recorded. HR (95% CI) 3.39 (1.57, 7.32)	Sarcopenic obese individuals had the highest risk of CVD mortality
[94]	A total of 664 consecutive cardiovascular surgery patients with a mean age of 65.8 ± 12.7 years	All-cause mortality as a prognosis	Psoas muscle attenuation (MA) and visceral adipose tissue (VAT) (abdominal obesity defined by WC)	EuroSCORE	HR (95% CI) 3.04 (1.25, 7.40)	Sarcopenic obesity was associated with all-cause mortality in patients undergoing cardiovascular surgery
[91]	UK Biobank/452 931 men and women aged 40 and 69 years 5.1 years follow-up	CHD events and CHD mortality, CVD events and CVD mortality	Handgrip strength Obesity = BMI >30 or waist to hip ratio ≥0.95 in men and ≥0.80 in women	Age, sex, ethnicity, smoking, alcohol consumption, diabetes, physical activity, deprivation	HR (95% CI) Sarcopenic obesity BMI CHD mortality NO history: 1.74 (1.26, 2.42) History: 1.53 (1.22, 1.93) CVD mortality NO history: 1.39 (1.07, 1.81) History: 1.14 (0.87, 1.48) Sarcopenic obesity WHR CVD mortality NO history of CVD: 2.04 (1.74, 2.38) History of CVD: 1.82 (1.55, 2.14)	Sarcopenic obese-associated CHD mortality irrespective of their CVD history CVD mortality associated only with those who did not have a history of CVD

ACEI: angiotensin-converting enzyme inhibitors; ARBs: angiotensin receptor blockers; BMI: body mass index; CHD: coronary heart disease; CVD: cardiovascular disease; DPP4: dipeptidyl peptidase 4; eGFR: estimated glomerular filtration rate; HOMA-IR: homeostasis model assessment of insulin resistance; WC: waist circumference; WHR: waist to hip ratio.

**Table 3 ijms-22-04339-t003:** Obesity (SO) definition, prevalence, cancer type, and stage among adults and older adults.

Author, Year, Country	Study Design	Local and Stage of Tumor	Age (*n*;% Sex)	SO Definition	Prevalence
Digestive System					
[105] Canada	Retrospective cohort	Gastrointestinal tract or respiratory system/I, II, III e IV (mostly III and IV)	35 to 88 years-old (*n* = 250; 54.4% male)	CT L3 SMI: ♀ ≤ 38.5 cm^2^/m^2^ and ♂ ≤ 52.4 cm^2^/m^2^; obesity: BMI ≥ 30 kg/m^2^	15%
[110] USA	Retrospective cohort	Colorectal/advanced	58 ± 12 years-old (*n* = 259; 60.0% male)	Sarcopenia TPA: ≤ 500 mm^2^/m^2^; obesity: BMI ≥3 0 kg/m^2^	2%
[24] The Netherlands	Retrospective cohort	Colorectal/advanced	24 to 86 years-old (*n* = 171; 60.8% male)	CT L3 SMI: ♀ < 41 cm^2^/m^2^ and ♂ < 43 cm^2^/m^2^ (BMI < 25 kg/m^2^) and < 53 cm^2^/m^2^ (BMI ≥ 25 kg/m^2^); obesity: % body fat > 44.4 ♀ and > 35.7 ♂	28.7%
[111] The Netherlands	Prospective cohort	Esophagus/I, II, III, and IV	63 ± 10 years-old (*n* = 123; 82.1% male)	CT L3 SMI: ♀ ≤ 38.5 cm^2^/m^2^ and ♂ ≤ 52.4 cm^2^/m^2^; obesity: BMI > 30 kg/m^2^ or visceral adiposity (L3)	2% (Sarcopenic obesity) 17% (Sarcopenic visceral obesity)
[107] Germany	Prospective cohort	Colorectal, cholangiocarcinoma, and liver/not specified	28 to 82 years-old (*n* = 80; 63.75% male)	CT L3 SMI: ♀ < 41 cm^2^/m^2^ and ♂ < 43 cm^2^/m^2^ (BMI < 25 kg/m^2^) and < 53 cm^2^/m^2^ (BMI ≥ 25 kg/m^2^); obesity: % body fat > 44.4 for ♀ and > 35.7 for ♂	22%
[112] Sweden	Retrospective cohort	Esophagus/I, II, III, and IV (mostly advanced)	47 to 83 years-old (*n* = 72; 85.0% male)	CT L3 SMI: ♀ ≤ 38.5 cm^2^/m^2^ and ♂ ≤ 52.4 cm^2^/m^2^; obesity: BMI ≥ 25 kg/m^2^	14%
[113] UK	Prospective database study	Colorectal/I, II, III, and IV	61 to 77 years-old (*n* = 805; 58.6% male)	CT L3 SMI: ♀ ≤ 38.5 cm^2^/m^2^ and ♂ ≤ 52.4 cm^2^/m^2^; obesity: BMI ≥ 30 kg/m^2^	10%
[32] China	Prospective study	Stomach/I, II, III, and IV	56 to 78 years-old (*n* = 636; 75.2% male)	Sarcopenia: L3 SMI ♂: ≤ 40.8 cm^2^/m^2^ and ♀: 34.9 cm^2^/m^2^; obesity: VFA ≥ 132.6 cm^2^ for ♂ and 91.5 cm^2^ for ♀	6.1%
[59] Korea	Cohort Study	Stomach (pre-cancerous and cancerous lesions)	49.1 ± 11.6 years (*n* = 8356; 55.9% male)	SMI estimated by ASM/body weight× 100 (%): 29.3% in ♂ and 26.7% in ♀; obesity: BMI ≥ 25 kg/m^2^	Pre-cancer: 14.6% Cancer 41.7%
[114]	Prospective study	nonmetastatic Rectal/I, II and III	59.0 ± 10.9 years (*n* = 1384; 64.2% male)	CT L3 SMI: ♀ ≤ 38.5 cm^2^/m^2^ and ♂ ≤ 52.4 cm^2^/m^2^; obesity: BMI > 25 kg/m^2^	22.2%
[98] Spain	Cohort Study	Stomach/I, II, and III (>50% III)	27 to 88 years-old (*n* = 198; 57.6% male)	CT L3 SMI: ♀ ≤ 38.5 cm^2^/m^2^ and ♂ ≤ 52.4 cm^2^/m^2^; obesity: VFA > 163.8 cm^2^ for ♂ and > 80.1 cm^2^ for ♀	28%
**Endocrine System**					
[115] Canada	Retrospective cohort	Pancreas/II to IV (mostly IV)	64.4 ± 9.3 years-old (*n* = 111; 53.2% female)	CT L3 SMI: ♀ ≤ 38.5 cm^2^/m^2^ and ♂ ≤ 52.4 cm^2^/m^2^; obesity: BMI ≥ 25 kg/m^2^	16.2%
[116] UK	Retrospective cohort	Pancreas/advanced	Palliative chemo: *n* = 98; 56.1% male; 64.8 ± 8.7 years; No chemotherapy: *n* = 130; 53.1% male; 72.9 ± 11.1	CT L3 SMI: ♀ < 41 cm^2^/m^2^ and ♂ < 43 cm^2^/m^2^ (BMI < 25 kg/m^2^) and < 53 cm^2^/m^2^ (BMI ≥ 25 kg/m^2^); obesity: BMI ≥ 30 kg/m^2^	25%
[117] Japan	Cross-sectional study	Liver/I, II, III, and IV	73.6 ± 7.8 years-old among sarcopenic obese	SMI ♂: 40.31 cm^2^/m^2^ and ♀: 30.88 cm^2^/m^2^; obesity: VFA ≥ 100 cm^2^	7%
[118] Japan	Retrospective study	Pancreas/not reported	68 years-old (*n* = 301; 55.8% male)	Sarcopenia: SMI ♂: 47.1 cm^2^/m^2^ and ♀: 36.6 cm^2^/m^2^; obesity: VFA ≥ 100 cm^2^	18.9%
[97] Austria	Retrospective analysis	Pancreas/III	34 to 87 years-old (*n* = 133; 51.1% male)	CT L3 SMI: ♀ ≤ 38.5 cm^2^/m^2^ and ♂ ≤ 52.4 cm^2^/m^2^; obesity: BMI ≥ 25 kg/m^2^	25.6%
**Genitourinary System**					
[119] Republic of Ireland	Retrospective study	Prostate/not reported	69 ± 8.3 years-old (*n* = 63; 100% male)	CT L3 SMI: ♀ < 41 cm^2^/m^2^ e ♂ < 43 cm^2^/m^2^ (BMI < 25 kg/m^2^) and < 53 cm^2^/m^2^ (BMI ≥ 25 kg/m^2^)	12.6%
**Integumentary System**					
[95] USA	Retrospective Cohort	Breast/I, II, and III	Not reported (*n* = 67; 100% female)	CT L3 SMI: ♀ ≤ 38.5 cm^2^/m^2^ and ♂ ≤ 52.4 cm^2^/m^2^; obesity: BMI ≥ 30 kg/m^2^	2.3%
[9] The Netherlands	Retrospective study	Breast/not reported	58.8 ± 11.3 years-old (*n* = 166; 100% female)	CT L3 SMI: ♀ < 41 cm^2^/m^2^ and ♂ < 43 cm^2^/m^2^ (BMI < 25 kg/m^2^) and < 53 cm^2^/m^2^ (BMI ≥ 25 kg/m^2^)	7.2%

SMI: skeletal muscle index; VFA: visceral fat area; ASM: appendicular skeletal muscle mass; ASMI: appendicular skeletal muscle index; TPA: total psoas muscle area; DXA: dual-energy X-ray absorptiometry; NHANES: National Health and Nutrition Examination Survey; CT: computerized tomography.

**Table 4 ijms-22-04339-t004:** Clinical implications and survival in adults and older adults with cancer and sarcopenic obesity.

Author, Year	Study Design and Sample	Implications on Clinic Features and Survival
Functional Outcome		
[105]	Retrospective cohort 35 to 88 y (*n* = 250; 54.4% male)	47.0% of individuals with SO presented decreased functional status
[106]	Cross-sectional study 64.5 ± 9.5 y (*n* = 28; 68.0% male)	Decreased handgrip strength in men
**Clinical and Surgical Outcome**		
[110]	Retrospective cohort 58 ± 12 y (*n* = 259; 60.0% male)	Prolonged hospitalization and higher postoperative complication rates, compared to non-sarcopenic
[95]	Retrospective cohort Age not reported (*n* = 67; 100% female)	Poorer prognostic compared to normal BMI.
[106]	Cross-sectional study 64.5 ± 9.5 y (*n* = 28; 68.0% male)	Men with SO showed more symptoms at Edmonton’s symptom assessment system (ESAS) compared to non-sarcopenic
[24]	Retrospective cohort 24 to 86 y (*n* = 171; 60.8% male)	Higher hospital readmission than non-SO
[112]	Retrospective cohort 47 to 83 y (*n* = 72; 85.0% male)	SO individuals showed higher toxicity risk during chemotherapy
[113]	Prospective database study 61 to 77 y (*n* = 805; 58.6% male)	SO increased postoperative complications compared to low muscle mass non-obesity along the same period
[120]	Prospective study (*n* = 206; 78.15% male)	Sarcopenia in overweight/obese individuals associated with a 6-fold increased postoperative risk of complications, higher hospital costs, and higher hospital readmission in 30 days after gastrectomy (compared to non-sarcopenic)
[96]	Retrospective analysis of a cohort 68 ± 10 y (*n* = 47; 68.1% male)	SO a risk factor for chemo-related toxicity in gastric cancer patients, leading to a premature stop
[97]	Retrospective analysis 34 to 87 y (*n* = 133; 51.1% male)	Higher complication rates
[32]	Prospective study 56 to 78 y (*n* = 636; 75.2% male)	Higher risk of postoperative complications
[119]	Retrospective study 69 ± 8.3 y (*n* = 63; 100% male)	SO not associated with chemo-related toxicity in castration-resistant metastatic prostate cancer
[59]	Cohort study 49.1 ± 11.6 y (*n* = 8.356; 55.9% men)	SO associated with gastric cancer diagnosis
[98]	Cohort study 27 to 88 y (*n* = 198; 57.6% male)	SO not associated with poorer postoperative recovery in gastric cancer patients
**Mortality/Survival Outcome**		
[105]	Retrospective cohort 35 to 88 y (*n* = 250; 54.4% male)	Shorter survival compared to non-sarcopenic.
[115]	Retrospective cohort 64.4 ± 9.3 y (*n* = 111; 53.2% female)	Survival 2.7-fold smaller in SO
[123]	Prospective cohort 42 to 81 y (*n* = 41; 56.0% female)	SO associated with a 4-fold higher risk of mortality
[116]	Retrospective cohort Palliative-chemo group (*n* = 98; 56.1% male): 64.8 ± 8.7 y No-chemo group (*n* = 130; 53.1% male): 72.9 ± 11.1	SO associated with smaller survival rates
[111]	Prospective cohort 63 ± 10 y (*n* = 123; 82.1% male)	SO not associated with postoperative mortality. Chemoradiotherapy-related muscle loss predicted postoperative mortality
[107]	Prospective cohort (*n* = 80; 63.75% male)	SO not associated with higher mortality
[124]	Retrospective cohort study 63 to 78 y (*n* = 262; 85.9% male)	SO not associated with mortality among bladder cancer patients
[96]	Retrospective cohort 68 ± 10 y (*n* = 47; 68.1% male)	SO associated with smaller survival rates among gastric cancer patients
[9]	Retrospective study 58.8 ± 11.3 y (*n* = 166; 100% female)	SO not associated with overall survival in women with metastatic breast cancer undergoing palliative care
[117]	Cross-sectional study 73.6 ± 7.8 y old among sarcopenic obese	Preoperative SO is an independent risk factor for death and recurrence of hepatocellular carcinoma in patients submitted to primary hepatectomy
[118]	Retrospective study 68 y (*n* = 301; 55.8% male)	SO associated with increased mortality
[97]	Retrospective analysis 34 to 87 y (*n* = 133; 51.1% male)	SO associated with shorter overall survival
[114]	Prospective study 59.0 ± 10.9 y (*n* = 1384; 64.2% male)	SO negatively associated with overall survival in nonmetastatic rectal cancer patients

SO: sarcopenic obesity.

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
