# Peer review of "The Role of Sarcopenic Obesity in Cancer and Cardiovascular Disease: A Synthesis of the Evidence on Pathophysiological Aspects and Clinical Implications"

_ijms, 2021, doi:10.3390/ijms22094339_

Round 1
Reviewer 1 Report
In this manuscript, the authors reviewed the role of sarcopenic obesity in cancer and cardiovascular diseases. Sarcopenic obesity is a relatively new concept and its role in cancer and CVD has not been widely investigated. Therefore, this review article will provide the fundamentals on this clinically important symptom, and is of great interest.
However, there are still unclear things. Especially, it is unclear whether sarcopenia and obesity occur in cooperation or independently. The relationship between sarcopenia and obesity should be discussed further. For example, how sarcopenia affects obesity and vice versa?
Also, basically, the phenotypes described in the CVD and cancer sections can occur just by either sarcopenia or obesity alone. It is not clear if both sarcopenia and obesity are necessary for any of the phenotypes.
In addition, the manuscript is somewhat difficult to read and also has errors in grammar and wordings including the title and abstract, which should be corrected.
Author Response
Professor Maurizio Battino
Editor-in-Chief
International Journal of Molecular Sciences
14 April 2021
Reviewer #1
In this manuscript, the authors reviewed the role of sarcopenic obesity in cancer and cardiovascular diseases. Sarcopenic obesity is a relatively new concept and its role in cancer and CVD has not been widely investigated. Therefore, this review article will provide the fundamentals on this clinically important symptom, and is of great interest.
Response:Thank you for your kind words. Much appreciated.
However, there are still unclear things. Especially, it is unclear whether sarcopenia and obesity occur in cooperation or independently. The relationship between sarcopenia and obesity should be discussed further. For example, how sarcopenia affects obesity and vice versa?
Response:Thank you for raising this important point. We have clarified this relationship in the revised manuscript as suggested. As the reviewer has pointed out, the concept of sarcopenic obesity is relatively new and, therefore, we have considered this condition as one rather than two independent conditions. In addition, we have included the following statement in the revised manuscript (please see lines 98-102): “Several of these outcomes are exacerbated when sarcopenia and obesity occur simultaneously rather than independently [15, 20]. Additionally, sarcopenic obesity is associated with a distinct profile of biomarkers which reflect reciprocal influence of adipose tissue on skeletal muscle and vice-versa [13, 20]”.
Also, basically, the phenotypes described in the CVD and cancer sections can occur just by either sarcopenia or obesity alone. It is not clear if both sarcopenia and obesity are necessary for any of the phenotypes.
Response:Thank you for your comment. Regarding cancer, all studies that we have included considered sarcopenic obesity as an outcome or main variable and not separate conditions. Based on the inclusion criteria of our literature review we only selected studies that analysed sarcopenic obesity as a condition. With regards to CVD, several studies have reported the associations between either sarcopenia or obesity alone. However, since the aim of our review was just sarcopenic obesity, we did not present the findings for sarcopenia or obesity separately. Sarcopenic obesity is a phenotype different from obesity or sarcopenia alone and as Atkins et al. (2014) showed, there was no interaction between sarcopenia and obesity in relation to CVD events, CVD mortality and all-cause mortality. This result was addressed in the revised version of the manuscript (please see lines 357-359).
In addition, the manuscript is somewhat difficult to read and also has errors in grammar and wordings including the title and abstract, which should be corrected.
Response:Thank you for your comment. The authors have revised the English language throughout the manuscript to address the issues raised by the reviewer in their comment.

Reviewer 2 Report
The authors have provided a through review of clinical studies published to date in the field of sarcopenic obesity on mortality in cancer and CVD.
(1) In the studies cited, please discuss if there is any evidence to suggest gender differences on the effect of sarcopenic obesity and CVD/cancer mortality. It seems fewer studies are done on women? If that is the case, please discuss. The authors indicate prevalence of increased sarcopenic obesity in endometrial, breast cancers etc, but what about theoverall effect of gender on mortality and Sarcopenic obesity.
(2) My main comment is with regards to the central figure 1. Please make sure the arrows are not in from of the text. Additionally it may be helpful if the effects of obesity are removed from the figure and it focuses only on sarcopenic obesity.
Furthermore please use a cartoon to show sarcopenic obesity vs. obesity and then elucidate the mechanism in greater detail on the existing figure. The details on information such as Irisin, adipokines and cytokines etc. exists in the text but please include in the figure. You can create small bullet points under every box.
(3) Needs a careful reading as several typos exist.
Example, line 250, 'two myosin', perhaps you mean myosin II,
Line 253, 'Irisina', maybe you meant Irisin,
line 127, 'smooth cells' should be 'smooth muscle cells'
(4) Please make further smaller subheadings where ever possible.
(5) The 'obesity paradox' is mentioned at the end of both the sections that increased BMI is protective for mortality from both CVD and cancer in certain cases. Perhaps create a common small subheading at the end for the explanation from line 519 to line 522. Include a couple more points or suggestions or explanation in it.
(6) Line 538 'specific preventative therapies to prevent sarcopenic obesity', cite a few examples of such therapies.
(7) Line 432, 'Muscle depletion can represent a modifiable risk factor in patients with colorectal 432 cancer and needs to be considered in such patients.' Please clarify, did you mean that muscle depletion is a modifiable risk factor specifically in colorectal cancer, if not please reword.
Author Response
We appreciate the careful assessment of our manuscript entitled “The role of sarcopenic obesity in cancer and cardiovascular disease: a synthesis of the evidence on pathophysiological aspects and clinical implications” by the reviewers and editorial board. The reviewers’ valuable contributions have improved considerably our manuscript.
All reviewers' comments were addressed in detail and the necessary changes were made. We have carefully considered every comment, promptly accepted all the suggestions, and made the alterations as recommended which are highlighted in red in the revised version of the manuscript. Please find below a point-by-point response to the Editors’ and Reviewers’ comments with our responses in blue font.
We appreciate the time and effort dedicated to the revision of our manuscript and we are at your disposal should any further corrections or changes be deemed necessary.
Yours sincerely,
Professor Erika Aparecida Silveira
Reviewer #2
The authors have provided a thorough review of clinical studies published to date in the field of sarcopenic obesity on mortality in cancer and CVD.
(1) In the studies cited, please discuss if there is any evidence to suggest gender differences on the effect of sarcopenic obesity and CVD/cancer mortality. It seems fewer studies are done on women? If that is the case, please discuss. The authors indicate prevalence of increased sarcopenic obesity in endometrial, breast cancers etc, but what about the overall effect of gender on mortality and Sarcopenic obesity.
Response:Thank you for raising this important point. We have found one study that analysed the association between sarcopenic obesity and gender in cancer patients. We have added this information in the revised Tables 3 and 4 (Please see reference 115: Han et al., 2020; lines 471; 493-497). With regards to CVD, the authors identified only one study that assessed the associations, stratified by gender, in which various cardiometabolic risk factors were more common in male and female individuals with sarcopenic obesity except for systolic blood pressure. These findings were included and discussed in the revised manuscript (please see lines 295-298).
(2) My main comment is with regards to the central figure 1. Please make sure the arrows are not in from of the text. Additionally, it may be helpful if the effects of obesity are removed from the figure and it focuses only on sarcopenic obesity. Furthermore, please use a cartoon to show sarcopenic obesity vs. obesity and then elucidate the mechanism in greater detail on the existing figure. The details on information such as Irisin, adipokines and cytokines etc. exists in the text but please include in the figure. You can create small bullet points under every box.
Response:Thank you for your comment. We have moved the arrows and cleaned the information on obesity, focusing only on sarcopenic obesity as suggested by the reviewer. Additionally, we have provided further information on molecular mediators of pathophysiology at the bottom of Figure 1.
(3) Needs a careful reading as several typos exist.
Example, line 250, 'two myosin', perhaps you mean myosin II, Line 253, 'Irisina', maybe you meant Irisin, line 127, 'smooth cells' should be 'smooth muscle cells'
Response:Thank you. We have corrected and re-written all the typos throughout the revised manuscript according to the reviewer’s suggestion comments.
(4) Please make further smaller subheadings wherever possible.
Response:Thank you. However, the authors believe that the existent headings and subheadings are clear enough and also didactic.
(5) The 'obesity paradox' is mentioned at the end of both the sections that increased BMI is protective for mortality from both CVD and cancer in certain cases. Perhaps create a common small subheading at the end for the explanation from line 519 to line 522. Include a couple more points or suggestions or explanation in it.
Response:Thank you. Additional explanation on the obesity paradox and CVD (please see lines 281-285) and cancer (please see lines 534-536) has been provided in the revised manuscript. The authors feel that having a new subheading, as suggested, will not improve the manuscript since it is not the main focus of the study. We have explained these aspects regarding both CVD and cancer.
(6) Line 538 'specific preventative therapies to prevent sarcopenic obesity', cite a few examples of such therapies.
Response:Thank you. The following statement has been included (please see lines 545-546) in the revised manuscript: “such as caloric restriction associated to protein supplementation and resistance exercise”.
(7) Line 432, 'Muscle depletion can represent a modifiable risk factor in patients with colorectal 432 cancer and needs to be considered in such patients.' Please clarify, did you mean that muscle depletion is a modifiable risk factor specifically in colorectal cancer, if not please reword.
Response:Thank you.We have rephrased the paragraph in question in the revised manuscript (please see lines 440-441) as follows: “Low lean body mass, mostly muscle, can represent a modifiable risk factor in patients with cancer, especially in non-advanced stages, and should be considered”.
